# Unsaturated Fatty Acid Synthesis Is Associated with Worse Survival and Is Differentially Regulated by *MYCN* and Tumor Suppressor microRNAs in Neuroblastoma

**DOI:** 10.3390/cancers16081590

**Published:** 2024-04-21

**Authors:** Dennis A. Sheeter, Secilia Garza, Hui Gyu Park, Lorraine-Rana E. Benhamou, Niharika R. Badi, Erika C. Espinosa, Kumar S. D. Kothapalli, J. Thomas Brenna, John T. Powers

**Affiliations:** 1Department of Pediatrics, Dell Pediatric Research Institute, Dell Medical School at The University of Texas at Austin, Austin, TX 78723, USA; dennis.sheeter@austin.utexas.edu (D.A.S.); hi.park@austin.utexas.edu (H.G.P.); lorrainerana.benhamou@austin.utexas.edu (L.-R.E.B.); niharikareddy.badi@austin.utexas.edu (N.R.B.); ece97@austin.utexas.edu (E.C.E.); 2Department of Chemistry, Dell Pediatric Research Institute, The University of Texas at Austin, Austin, TX 78723, USA; seci@utexas.edu; 3Department of Nutritional Sciences, College of Natural Sciences, The University of Texas at Austin, Austin, TX 78712, USA; ksdkumar@gmail.com; 4Division of Pharmacology and Toxicology, College of Pharmacy, The University of Texas at Austin, Austin, TX 78712, USA

**Keywords:** neuroblastoma, *MYCN*, *MYC*, Warburg effect, unsaturated fatty acids, microRNA, arachidonic acid

## Abstract

**Simple Summary:**

Cellular use of important fuel sources like glucose is disrupted in most human cancers. The MYC family of pro-cancer proteins is known to contribute to these shifts in metabolism. Relatively little is known about any such shifts in fatty acid use, which is unfortunate because fatty acids, particularly unsaturated fatty acids, are rich fuel sources and can also serve important roles in cellular survival and immune system function. Here, we aimed to better understand the role of unsaturated fatty acid synthesis in pediatric neuroblastoma, a devastating disease in children that is driven by the MYC family member MYCN, where less than half of high-risk patients survive. We discovered that the levels of omega-3, omega-6, and omega-9 unsaturated fatty acids and the genes that synthesize them are highly disturbed in aggressive neuroblastoma. We further connected these disrupted patterns to MYCN and microRNAs, a special class of small RNAs important in neuroblastoma and other cancers. This work provides new connections between fatty acid synthesis and poor outcomes in neuroblastoma, providing a better understanding of how this disease may use fatty acids as it develops and progresses and some potential new considerations for treatment.

**Abstract:**

*MYCN* amplification (*MNA*) and disruption of tumor suppressor microRNA (TSmiR) function are key drivers of poor outcomes in neuroblastoma (NB). While MYCN and TSmiRs regulate glucose metabolism, their role in de novo fatty acid synthesis (FAS) and unsaturated FAS (UFAS) remains poorly understood. Here, we show that FAS and UFAS (U/FAS) genes *FASN*, *ELOVL6*, *SCD*, *FADS2*, and *FADS1* are upregulated in high-risk (HR) NB and that their expression is associated with lower overall survival. RNA-Seq analysis of human NB cell lines revealed parallel U/FAS gene expression patterns. Consistent with this, we found that NB-related TSmiRs were predicted to target these genes extensively. We further observed that both MYC and MYCN upregulated U/FAS pathway genes while suppressing TSmiR host gene expression, suggesting a possible U/FAS regulatory network between MYCN and TSmiRs in NB. NB cells are high in de novo synthesized omega 9 (ω9) unsaturated fatty acids and low in both ω6 and ω3, suggesting a means for NB to limit cell-autonomous immune stimulation and reactive oxygen species (ROS)-driven apoptosis from ω6 and ω3 unsaturated fatty acid derivatives, respectively. We propose a model in which MYCN and TSmiRs regulate U/FAS and play an important role in NB pathology, with implications for other MYC family-driven cancers.

## 1. Introduction

NB is a highly metastatic pediatric cancer derived from the neural crest-sympathoadrenal lineage [1]. While NB represents only 6% of childhood cancer incidence, it causes over 15% of childhood cancer deaths. Among NB patients, prognosis is strongly linked to risk group classification. Risk group assessment is defined by both clinical and biological variables. Risk grouping, which is modeled after the clinical trials of the Children’s Oncology Group Neuroblastoma Studies, is based on the combination of tumor stage defined using the International Neuroblastoma Risk Group Staging System staging guidelines, age at diagnosis, *MNA* status, loss of heterozygosity at 1p or 11q, mitotic karyorrhectic index (MKI, a measure of both mitotic and fragmented nuclei cells within a tumor), and histopathological classification (a combination of neuroblastoma vs. ganglioneuroblastoma status, differentiation subtype, Schwannian stroma content, MKI, and age at diagnosis). Appropriate identification of HR and LR patients is an ongoing effort to classify patients most effectively within differential groups of expected prognoses.

In addition to the overarching risk categorization of neuroblastoma, further patient stratification occurs based on distinct genetic patterning within risk groups. For example, high-risk neuroblastoma is further subdivided by the presence or absence of *MNA*. *MYCN* is the defining oncogene of NB and is part of a small family of proto-oncogenic transcription factors that includes *MYC* and *MYCL*. *MNA* is a hallmark of HR disease and poor patient outcomes, where overall patient survival is below 50% [2,3,4]. *MYCN* overexpression, outside of its essential roles in embryonal and neuronal development, drives increased cellular proliferation and contributes to tumorigenesis. *MYCN* status is central to NB genetic patterning and is strongly correlated with the presence of other risk-status defining chromosomal alterations, such as chromosomal deletions of 1p and 11q. It is critical to note that while *MNA* is a dominant genetic event in NB, it represents only half of all HR disease. *MYCN* non-amplified (*MNon*) HR disease has a poor prognosis similar to *MNA*, despite differential contribution of *MYCN* between the two groups. In some cases, *MYC* is overexpressed in *MNon* HR disease. In others, different genetic drivers such as *ATRX* mutation and chromosome 3p and 11q loss can play dominant roles in *MNon* HR pathology. Additionally, microRNAs have repeatedly been shown to play a central role in both *MNA* and *MNon* HR NB. Both *MYCN* and *MYC* modulate microRNA expression and function and are themselves sensitive to microRNA regulation. The interplay between NB risk class, *MYCN* status, and microRNA function is part of the central message of this work.

In contrast, lower risk tumors have favorable overall survival (90–95%), whereas HR disease is notoriously difficult to treat and has an overall survival (OS) rate below 50%. Consequently, high-risk neuroblastoma patients endure one of the most intensive genotoxic therapeutic regimens in all of oncology. Children that do survive are often left with life-long complications and increased risk for secondary cancers. Consequently, increased understanding of the genetic contributors to HR disease will aid in the development of more effective and less genotoxic treatments for HR patients.

Both *MYCN* and *MYC* (*MYC/N*) are *MYC* family transcription factors with highly overlapped function [5,6]. These genes play important roles in regulating cellular growth and metabolism, including glycolysis and lipid synthesis. For example, MYC overexpression drives the Warburg effect, where glucose metabolism shifts towards rapid generation of ATP via anaerobic glycolysis over oxidative phosphorylation (Figure 1a). Multiple genes in this pathway are known transcriptional targets of MYC, which has also been implicated in the expression of U/FAS genes *ACLY*, *ACACA*, *FASN*, and *SCD* (Figure 1b) [7,8]. Together with *ELOVL6*, this pathway usually terminates with the synthesis of oleic acid (OA) when sufficient polyunsaturated fatty acids are available [9]. Similarly, *MYCN* is associated with enhanced glycolysis and metabolic reprogramming and has also been implicated in fatty acid uptake and synthesis in NB [10,11,12].

Gene expression is highly regulated at both transcriptional and post-transcriptional levels. MicroRNAs (miRNAs) are small, ~22 nucleotide-long single-stranded RNAs that post-transcriptionally decrease mRNA stability and/or translation efficiency through sequence-specific binding to the 3′ untranslated regions (3′ UTRs) of target mRNAs. MiRNAs play roles in most cellular activities, including multiple metabolic activities such as the regulation of glucose metabolism, obesity, and fatty acid oxidation by *let*-7, *mir*-22, and *miR*-33, respectively [13,14,15]. MiRNAs have also been implicated throughout the steps of glycolysis (Figure 1a) [16,17]. MiRNAs also play both oncogenic and tumor-suppressive roles in cancer [18]. Altered miRNA function is key to NB development, including a central role for the impairment of the *let-7* tumor suppressor miRNA (TSmiR) family [19,20,21,22]. Additional NB-associated TSmiRs have been identified, including *miRs*-*22*, *-101*, *-124*, *-140*, *-150*, and *-204* [20,23,24,25,26,27]. Little is known about miRNA control of UFAS.

In addition to their role as energy substrates, FAs are the main structural components of all cellular membranes. Specific non-esterified free fatty acids are key to intracellular and paracrine cell signaling, as well as when acylated to phosphatidylinositol and diacylglycerol. The location of the double bonds that define a fatty acid as unsaturated are measured from the terminal methyl group; they also define three important classes of unsaturated fatty acids: omega-9 (ω9) and the essential FAs ω6 and ω3 (Figure 1b, Appendix A) [28].

Highly unsaturated FAs (HUFAs), such as ω6 arachidonic acid (ARA, 20:4), serve as a precursor for numerous bioactive oxylipins, such as the immune-stimulating and pro-inflammatory molecules leukotrienes, prostaglandins, and 20-HETE [29]. The ω3 HUFA docosahexaenoic acid (DHA, 22:6) is critical for normal neuronal health and induces apoptosis in multiple cancer cell types [30,31,32]. Ratios of fatty acyl chains within membrane phospholipids influence the signaling molecules produced on an ‘as needed’ basis in response to stimulation [33]. With respect to cancer, fatty acid profiles can be a passenger or contribute causally; transformation to malignancy alters the fatty acids of membranes, and fatty acid inputs are known to influence tumorigenesis [34,35,36,37,38,39,40,41,42].

In this study, we aim to understand the expression patterns and outcome associations of saturated and unsaturated fatty acid synthesis in NB, as well as fatty acid levels in vitro in NB cell lines derived from HR tumors. Using complementary analysis of publicly available genomic datasets and cultured NB cell measurements, we examined the potential regulation of UFAS genes through MYCN and MYC activity and post-transcriptional regulation by TSmiRs. Finally, we profiled the levels of ω3, ω6, and ω9 unsaturated FAs in human NB cell lines. A consistent pattern emerged between HR NB outcomes, UFAS gene expression and regulation, and ω9 vs. ω3 and ω6 HUFA lipid profiles, providing a novel organizing principle to fully understand how the regulation of fatty acid synthesis and metabolism may influence NB pathology, with implications for other MYC family-driven cancers.

## 2. Materials and Methods

Cell Lines: The following cell lines were used in this study: NHF (BJ, RRID CVCL_3653), BE1 (SK-N-BE(1), RRID CVCL_9898), BE2 (SK-N-BE(2), RRID CVCL_0528), C15 (CHLA-15, RRID CVCL_6594), C20 (CHLA-20, RRID CVCL_6602), KAN (SMS-KAN, RRID CVCL_7131), KANR (SMS-KANR, RRID CVCL_7132), KCN (SMS-KCN, RRID CVCL_7133), KCNR (SMS-KCNR, RRID CVCL_7134). All NB cell lines were obtained from the Childhood Cancer Repository at Texas Tech University Health Science Center, which is part of the Children’s Oncology Group. BJ normal human fibroblasts (NHFs) were purchased from ATCC. All cells were grown in RPMI-1640 medium supplemented with 10% fetal bovine serum and were tested for mycoplasma before and during use. Details for all the cell lines used in this study are shown in Appendix A.

RNA-Seq: Total RNA was submitted to LC Sciences (Houston, TX, USA) for library prep and sequencing. Sequencing (without spike-ins) was performed at a depth of 40 million 150 bp paired-end reads. Reads were normalized as Fragments Per Kilobase of transcript per Million mapped reads (FPKM). In the present study, we report UFAS genes and will report other genes elsewhere. RNA-Seq data sets are (will be) available under the GEO reference series GSE (http://www.ncbi.nlm.nih.gov/geo/query/acc.cgi?acc=GSE(TBD) (accessed on 25 March 2024)). Two batches of NB cell lines were subjected to RNA-Seq analysis; Batch 1: NHF, BE1, and BE2 (three biological replicates per cell line); Batch 2: NHF, C15, C20, KAN, KANR, KCN, and KCNR (two biological replicates per cell line).

Fatty Acid Profiling: Fatty acid analysis of cultured cells was performed as described previously [43,44,45,46]. Briefly, cell pellets were harvested after centrifugation, and fatty acid methyl esters were prepared according to a modified one-step method (three biological replicates per cell line) [47]. Fatty acid methyl esters were structurally identified by gas chromatography (GC)–chemical ionization mass spectrometry [48,49] and quantified using a GC-flame ionization detector after calibration with an equal-weight fatty acid methyl ester standard mixture to develop response factors. The statistical tests used are shown in each figure legend. Statistical significance was set at *p* < 0.05.

MicroRNA target site identification: The TargetScan platform (version 8.0) was used to identify predicted miRNA target sites within the 3′ UTR for each gene using the RefSeq sequence from transcript variant 1 [50]. For each gene, TargetScan lists all predicted miRNA sites within the 3′ UTR region of a transcript pertaining to a single Ensemble ID (ENST). For some of the genes, the Ensembl 3′ UTR transcript analyzed by TargetScan was longer than that found within the RefSeq sequences for a gene’s transcript variant 1. We adhered to the RefSeq 3′ UTR length (Figure 2). Briefly, 3′ UTRs were matched to the predicted binding of TSmiRs known to be associated with NB, specifically *miR-1*, *let-7*, *miR-22*, *miR-24*, *miR-34*, *miR-101*, *miR-124*, *miR-125*, *miR-140*, *miR-150*, and *miR-204*.

Secondary analysis: The SEQC-498 human NB patient dataset from “An Investigation of Biomarkers Derived from Legacy Microarray Data for Their Utility in the RNA-Seq Era” was used for human NB patient analysis [51]. Gene expression graphs, Cox regression analysis, Kaplan–Meier curves, and KEGG pathway analysis of human NB patients were generated using the data analysis tool suite on the r2: Genomics Analysis and Visualization Platform [52]. HR NB patients in this peer-reviewed study were identified as patients with stage 4 disease, >18 months at diagnosis, and patients of any age and stage with *MNA* tumors [51]. Blue-yellow z-score heatmaps in the main and supplemental figures were generated from tools on the R2 platform and are a visual representation of standard deviations above or below the average expression of the entire patient population for a given gene in each patient’s tumor. Using z-scores in these analyses allows for collective relative comparison of enriched gene expression among different genes, between different patient samples, and even within a single patient by measuring variance from the mean as opposed to raw expression values. Other datasets analyzed on the R2 database include the following: for multi-dataset comparison of U/FAS gene expression in NCC and NB cell lines: (Van Groningen et al., 2017 [53]) (r2 internal identifier: ps_avgpres_gsenatgen2017geo52_u133p2); for analysis of MYCN and MYC expression across NB patient datasets: control tissues, Adrenal Gland (various) (R2 internal identifier: ps_avgpres_adrenalglandns13_u133p2), Etchers [54] NB datasets, Lastowska [55], Delattre (r2 internal identifier: ps_avgpres_gse14880geo34_u133p2), and Versteeg [56]; for expression analysis of U/FAS and MYCN-regulated genes: MYCN inhibition by the JQ-1 bromodomain inhibitor in BE(2)C NB cells [57]; for analysis of miRNA expression in TH-MYCN murine NB: “The role of miRNAs in NB tumor development” (r2 internal identifier: mir_avgpres_thmycn24_mirbase19mm2); for Kaplan–Meier curves and additional NB dataset results (Appendix A) (Kocak et al., 2013 [58]). A full list of included datasets is provided in Appendix A.

Additional datasets from other sources include the following: for analysis of U/FAS gene expression changes in response to TSmiR modulation: doxycycline induced *miR-22* expression in RD18 rhabdomyosarcoma cells [59], *miR-22* transfection in 293T human embryonic kidney cells [60], *LIN28A* knockdown in H9 human embryonic stem cells, which causes inhibition of LIN28A resulting in a significant increase in *let-7* family members [61,62], *let-7* transfection in human dermal lymphatic endothelial cells (public on GEO on 23 February 2023; not yet published), and PEO1 *miR-124* transfection of PEO1 ovarian cancer cells [63]; for gene expression analysis of U/FAS- and MYC/N-regulated genes: tetracycline “off” *MYC* induction in P493-6 human EBV-transformed B lymphocytes [64]; for the adrenal gland and tibial nerve sample analysis (Appendix A): data were obtained from the r2 database using GTEx Portal data, (v8, protein coding), dbGaP accession number phs000424.vN.pN. Adrenal gland and tibial nerve samples from ages 20–49 were used in our analysis.

## 3. Results

To investigate the potential involvement of U/FAS genes in NB, we performed KEGG pathway analysis to identify metabolic pathways enriched in HR over low-risk (LR) NB patients. We observed significant enrichment of the UFAS pathway in HR disease. The de novo FAS pathway was also elevated, although not significantly (Figure 3a). Z-score-based heatmap analysis of the FAS and UFAS genes showed a clustered HR patient group. where UFAS genes *SCD*, *FADS2*, and *FADS1* as well as the de novo FAS genes *FASN* and *ELOVL6* are present at levels several standard deviations above the average expression level in the total patient pool (Figure 3b). In contrast, *ELOVL5* and *ELOVL2* were not enriched in these same patients. These genes comprise most of the U/FAS genes outlined in parallel fatty acid metabolism pathways (Figure 1b), suggesting that the majority of U/FAS genes may contribute to an important genetic circuit in HR disease. 

We next analyzed the relationship between U/FAS gene expression and overall patient survival; we performed Cox regression analysis of U/FAS pathway genes in two different NB patient studies. Expression of *FAS*, *ELOVL6*, *SCD*, *FADS2*, and *FADS1* was associated with elevated risk of death in NB patients from both studies, whereas *ELOVL5* and *ELOLV2* expression was associated with reduced risk (Figure 4a). Quantified expression differences of U/FAS pathway genes (Figure 1b) in HR vs. LR NB patients were consistent with these hazard ratios, revealing significant upregulation of all U/FAS genes except *ACACA*, *ELOVL5*, and *ELOVL2* in HR disease (Figure 4a), which is consistent with the HR patient gene enrichment clusters and Cox regression analysis (Figure 2b and Figure 3a). These data are also consistent with U/FAS gene expression-based Kaplan–Meier analysis of patient survival in both studies (Appendix A). We also examined the expression patterns of this gene set in eight human NB cell line pairs derived from tumors of the same patient at diagnosis (BE1, CHLA15, KAN, KCN) and post-relapse (BE2, CHLA20, KANR, KCNR) using RNA sequencing (RNA-Seq) analysis (Appendix A). We observed a similar pattern of U/FAS gene upregulation in NB cells compared with non-neural yet genetically intact NHF (Figure 4c). While CHLA15 and CHLA20 are *MNon* NB cell lines that show a similar pattern of gene expression to the *MNA* lines, they are known to have elevated *MYC* expression, which is consistent with our own RNA sequencing analysis (Appendix A). Our data are thus consistent with previous reports and support the idea that MYCN and MYC can contribute to elevated expression of the U/FAS genes of interest. Given that NB emerges from the neural crest developmental lineage, we next compared expression of U/FAS genes in human NB cell lines (*n* = 22) to human neural crest cells (NCC, *n* = 5) through secondary analysis of microarray data. We again observed that U/FAS genes *FASN*, *ELOVL6*, *SCD*, *FADS2*, and *FADS1* were upregulated in NB compared to NCC, whereas *ELOVL5* and *ELOVL2* were not significantly changed (Figure 4d).

The distinct expression and overall survival patterns associated with these genes suggest that U/FAS synthesis and metabolism may play an important role in the pathology of HR NB. We therefore examined the fatty acid profiles of NHF and eight NB cell lines that were grown under identical growth conditions using gas chromatography and mass spectrometry (Appendix A). NB cells displayed significantly reduced levels of ω3 fatty acids DHA (22:6ω3) compared to NHF, while ALA (18:3ω3) and EPA (20:5ω3) were essentially absent in NB (Figure 5a). Ω6 fatty acids LA (18:2ω6), ARA (20:4ω6), and AdrA (22:3ω6) were all reduced in NB, except for KCN cells, where AdrA levels were similar to NHF (Figure 5b). Notably, levels of ω9 Mead acid (20:3ω9) and its elongation product, docosatrienoic acid (DTrA, 22:3ω9), were dramatically increased beyond levels seen in normal cells. De novo monounsaturated fatty acid OA (18:1ω9) made up the highest percentage of fatty acids for all cell types, but trended higher in NB cells (Figure 5c).

Elevated Mead acid (20:3ω9), normalized to arachidonic acid (ARA, 20:4ω6), known as the triene–tetraene (T:T) ratio, is the accepted biochemical index of essential fatty acid deficiency (EFAD) [65]. In plasma, a T:T ratio greater than 0.4 indicates EFAD; a similar threshold applies to cells [66]. In NB cell lines, ARA levels decreased from 10% *w*/*w* in NHF to ~5% *w*/*w*, while Mead acid increased from trace levels to 2–5% of total fatty acids, reflecting EFAD for each NB cell line (Figure 5d,e). Plots of ARA and Mead acid levels indicate a negative correlation, while ARA and its elongation product, AdrA, indicates positive correlation, as expected (Figure 5f). The sum of Mead acid and ARA was 8.8% ± 1.2%, *w*/*w* (mean ± SD), reflecting the compensatory increase in Mead acid levels as ARA decreased (Figure 5g) [67].

Normally, EFAD arises because of the restricted supply of the ω6 essential fatty acid linoleic acid (*LA*, 18:2ω6) (Figure 1b, Appendix A) [68]. However, NHFs grown in the same media as the NB cells had only trace amounts of Mead acid and therefore are not EFAD. The enhanced expression of U/FAS genes in NB (Figure 4a,c,d) suggests possible directed synthesis of ω9 Mead acid. KEGG pathway analysis of FAS pathways in HR and LR NB patients revealed that while the de novo FAS pathway genes were uniquely enriched in HR patients, the majority of ω3 precursor α-linolenic acid (ALA, 18:3ω3) and ω6 precursor LA metabolism genes, represented by UFAS, were suppressed (Figure 5h). In contrast, genes from the three pathways displayed no discernable pattern in both the normal adrenal gland and tibial nerve tissue samples (Figure 5h, Appendix A), suggesting that a genetic program favoring de novo ω9 FAS over ω6/ω3 UFAS may be selected for in HR NB. 

HUFAs serve as precursors to eicosanoids, a subset of oxylipins and other locally active bioactive signaling lipids. Derivatives of ω6 ARA are produced via genes from three main pathways: lipoxygenase (*ALOX5*), cyclooxygenases (*COX2*, *PTGDS*, and *TBXAS1*), and CYP450 enzymes (Figure 6a) [29]. Of the more than 50 unique CYP450 enzymes, 15 are epoxygenases that act on HUFAs such as ARA. Only two of these 15 are expressed in NB: *CYP2C8* and *CYP4V2* (Appendix A). ARA processing by these enzymatic arms produces several pro-inflammatory eicosanoids, including prostaglandins, thromboxanes, leukotrienes, EETs, and 20-HETE [69]. *ALOX5* is the most abundant of the six human lipoxygenases in NB and has been implicated as a tumor suppressor (Appendix A) [70]. Cox regression analysis expression of these ARA metabolizing genes revealed uniform association of worse overall survival and low gene expression in two independent studies of NB patient survival (Figure 6b). They are also collectively downregulated in HR NB compared to LR patients (Figure 6c), suggesting a possible selective advantage for both reduced ω6 levels and ARA derivatives in NB. Given the importance of *MYCN* in NB, we next determined if MYCN can mediate expression of these genes. In both an inducible MYCN model using SK-N-AS cells (Bandino et al., 2014 [68]) and the tyrosine hydrolase-MYCN (*TH-MYCN*) mouse model of NB driven by a human *MYCN* transgene, we observed that *MYCN* expression resulted in reduced levels of *ALOX5*, *COX2*, *EPHX2*, and *CYP4V2* (murine *Cyp4v3*) (Figure 6d), suggesting that MYCN may play a role in ARA metabolism.

Overexpression of both *MYCN* and *MYC* drives metabolic conversion to favor the rapid generation of ATP via anaerobic glycolysis over oxidative phosphorylation [9,11,71]. While *MYCN* is the dominant oncogene in NB, MYC has been reported to play a role in some *MNA* tumors [6]. Given the strong overlap in MYCN and MYC functions, we next examined the relative expression of *MYCN* and *MYC* in both *MNA* and *MNon* disease in three distinct NB patient studies. *MYCN* exhibited the expected high expression pattern in *MNA* disease across all three patient datasets, displaying remarkable consistency in expression levels in both *MNA* and *MNon* tumors (Figure 7a,b). *MYC* also displayed stable but lower expression patterns across the datasets, revealing a reciprocal expression pattern with *MYCN*. When considered together, total *MYC/N* expression levels were higher in all patients from all three studies than in tissues originating in the adrenal gland and trunk neural crest, demonstrating that even *MNon* tumors have an elevated *MYC/N* signature. *MNA* patients had the highest *MYC/N* dose, as expected, although the magnitude of *MYCN* increase over *MNon* is likely somewhat mitigated by contributing yet reciprocal *MYC* expression. This cooperative pattern is consistent with MYCN and MYC having similar effects on metabolic reprogramming. KEGG pathway analysis of altered metabolic or cancer-related pathways in *MNA* and *MNon* NB patients identified the UFAS and FAS pathways as overrepresented in *MNA* patients (Figure 7c).

Given the predominance of *MYC/N* levels in *MNA* NB and the strong connection between the *MYC* family and metabolic reprogramming, we hypothesized that both *MYC* and *MYCN* would also influence U/FAS gene expression. Therefore, we analyzed U/FAS gene expression in complementary datasets of induced *MYC* expression in P493-6 lymphoid cells (Figure 7d) and JQ-1 bromodomain inhibitor-mediated MYCN inhibition in BE(2)C NB cells (Figure 7e) [71]. As previously shown, *MYC* induction resulted in a significant upregulation of the U/FAS genes *ACACA* (6.6×), *FASN* (9.2×), *SCD* (33.2×), and *ELOVL6* (2.6×). *FADS2*, *ELOVL5*, and *FADS1* expression levels were also significantly increased by 2.6×, 1.6×, and 3.1×, respectively. Known MYC-activated (*miR-17hg* and *CDC45*) and -suppressed (*LAMP2*, *p21*, and *p27*) genes behaved as expected. The host genes for *miR-22*, *miR-24*, and *miR-101* were also suppressed, suggesting that MYC may directly inhibit their transcription and may result in lower levels of these miRNAs in NB (Figure 7d).

In BE(2)C cells treated with the bromo-domain inhibitor JQ-1, known to downregulate *MYC/N* expression, U/FAS genes were consequently downregulated between 0.4× and 0.67× (Figure 7e). *FADS2* and *ELOVL6* were the most affected genes, with expression decreasing to 33% of the control in both cases. MYC family target genes once again behaved as expected, with decreased *miR-17hg* and *CDC45* levels and de-repressed *LAMP2* and *p21* expression. Consistent with *MYC* induction, inhibition of MYCN resulted in increased levels of host genes for *miRNA-22*, *miRNA-24*, and *miRNA-101*, further supporting a repressive role of MYC/N in their regulation.

Conversion from oxidative phosphorylation towards lactate production is common in cancer [72]. Post-transcriptional regulation of glycolysis pathway genes by multiple TSmiRs contribute to this conversion [16,73]. Therefore, we reasoned that neuroblastoma TSmiRs might similarly target U/FAS pathway genes [74,75]. We then analyzed the 3′ UTRs of U/FAS genes for the presence of predicted miRNA binding sites for a group of NB-associated TSmiRs: *miR-1*, *let-7*, *miR*-*22*, *miR-24*, *miR-34*, *miR-101*, *miR-124*, *miR-125 miR-140*, *miR-150*, and *miR-204* [20,23,24,25,26,27]. We observed an average of 10.9 target sites within the 3′ UTRs of U/FAS genes (average of 3.8 sites per kb of 3′ UTR), suggesting a possible unified post-transcriptional regulation of U/FAS genes by TSmiRs. The MYC family displayed an average of 5.3 TSmiR sites per 3′ UTR (3.1 sites per kb), with *MYCN* itself containing an extraordinary nine TSmiR sites in its 3′ UTR, despite its relatively short length (Figure 2, Appendix A). In contrast, housekeeping genes *SDHA*, *ACTB*, and *GAPDH* have minimal TSmiR sites (Figure 2). Due to their short 3′ UTRs (ranging from 201 to 662 nt), we also included three non-U/FAS-related genes with long 3′ UTRs. Taken together, these non-U/FAS genes contain an average of 2.3 TSmiR sites per 3′ UTR (0.9 sites per kb). These ratios of TSmiR sites suggest targeted regulation of the U/FAS genes by TSmiRs, similar to what has been reported for glycolysis (Figure 1a). Several NB-related TSmiRs, including *let-7*, *miR-124*, and *miR-204*, have also been linked to glycolysis gene regulation (Figure 1a) [16,20,25,27,73,76], suggesting that a TSmiR regulatory network may influence both glycolysis and U/FAS.

The TSmiRs assessed here have been previously demonstrated to possess anti-growth, tumor-suppressor functions and frequently reduced expression in NB [74,75]. We observed their collective downregulation in the *TH-MYCN*-driven murine NB compared to the wild-type sympathetic ganglia tissue of origin, demonstrating broad inhibition of multiple TSmiRs during NB development and collectively confirming previous reports of impaired TSmiR expression and function in NB (Figure 8a) [74,77,78]. The oncogenic *miR-17-92* cluster found in *miR-17hg*, which includes miRNAs *miR-17*, *miR-18a*, *miR-19a*, *miR-19b*, *miR-20a*, and *miR-92a* is a known transcriptional target of MYCN and is upregulated in *TH*-*MYCN* tumors, as expected.

A recent meta-analysis of miRNA-mRNA inhibition mechanisms revealed that at least half of all miRNA target genes are primarily regulated through mRNA decay [79]. We thus reasoned that mRNA levels of U/FAS TSmiR target genes may be reduced by elevated TSmiR activity; therefore, we examined the effects of TSmiR overexpression on *miR-22*, *let-7*, and *miR-124* targeted U/FAS gene mRNA levels in existing TSmiR overexpression datasets (Figure 8b). Fold-change heatmaps of two independent *miR-22* overexpression studies displayed similar patterns of reduced expression of *miR-22* targets *SCD* and *ELOVL6*, which contains four miR-22 sites in its 3′ UTR (Figure 2) and was strongly suppressed in both studies. In contrast, the expression levels of both *FADS2* and *ELOVL5*, which are not targeted by *miR-22*, were not affected (Figure 8b, left panel). A similar pattern of reduced expression was observed for *let-7* targets *SCD*, *FADS2*, and *MYCN* in studies of de-repressed *let-7* biogenesis and *let-7* transfection. Levels of non-*let-7* targets *FADS1*, *ELOVL6*, and *ELOVL5* were relatively unaffected, whereas non-*let-7* target *FASN* levels were reduced in the *let-7* biogenesis study but minimally affected by transfected *let-7* (Figure 2; Figure 8b, middle panel). *MiR-124* transfection resulted in the reduction of *miR-124* targets *SCD*, *FADS1*, and *ELOVL5*, whereas the non-*miR-124* targets *FADS2*, *FASN*, and *ELOVL6* showed no decrease (Figure 2; Figure 8b, right panel).

MiRNA genetic loci are often located within the introns of larger host genes and are frequently coordinately transcribed as part of the host transcript from which they are processed [80,81]. Thus, miRNA host gene expression patterns can serve as a surrogate for intragenic miRNA expression levels when small RNA sequencing data or qPCR analyses are not available. Therefore, we compared the expression levels of *miR-22hg* and *miR-101hg* to the expression levels of the individual *miR-22* targets *ELOVL6* and *SCD* as well as the *miR-101* targets *FASN*, *FADS2*, *ELOVL2*, and *MYCN* (Figure 8c). Both *miR-22* target genes were inversely correlated with the expression of the miRNA host genes. *FASN*, *FADS2*, and *MYCN* were inversely correlated with *miR-101hg* expression, whereas *ELOVL2* expression was not significantly correlated. These expression patterns provide further support for a model in which U/FAS genes are broadly targeted by TSmiR in NB. While these results are consistent with the targeting of FAS genes by TSmiRs, further work is required to validate these observations.

## 4. Discussion

*MYC*, *MYCN*, and numerous TSmiRs have been implicated in glucose metabolism. We propose that the MYCN/TSmiR regulatory program may similarly regulate the U/FAS pathway in NB through elevated expression of U/FAS genes *FASN*, *ELOVL6*, *SCD*, *FADS2*, and *FADS1* in HR disease, all of which are themselves associated with worse overall survival (Figure 1, Figure 2 and Figure 3). High levels of ω9 Mead acid and DTrA in cultured NB cells and concordant gene expression data between cells and NB patients suggest that the upstream FAS pathway consisting of *FASN* and *ELOVL6* can initiate the U/FAS pathway for the directed production of ω9 unsaturated FAs. Careful inspection of the ω3 fatty acids showed that only terminal ω3 DHA was present at an appreciable concentration, with precursor ALA and intermediate ω3 EPA at trace levels (Figure 5a). Similarly, for ω6 fatty acids, the LA precursor is low in all cells, while the usual terminal product ARA and its elongation product AdrA are present (Figure 5b). Upregulation of the U/FAS genes and the upstream pathway may induce a superficial EFAD via unusually high production of Mead acid (Figure 5c–e).

While assessments of ω9 Mead acid in cancer are limited, it was shown to be elevated in the serum and tumors of 38% of patients with hepatocellular carcinoma. It was not present in other tissues, suggesting that Mead acid production was elevated only in the tumor itself [41]. Mead acid treatment increased proliferation of HRT-18 breast cancer cells, whereas ω3 EPA (Figure 1b, Appendix A) reduced cell growth [82]. Furthermore, Mead acid increases both invasion activity and cell growth in multiple squamous cell cancer lines [83]. Together, these studies along with our observations in NB cells collectively suggest that Mead acid may provide a growth advantage in cancer, which could provide a selective rationale for elevated ω9 Mead acid production in NB.

*FADS2* dysregulation at the 11q13 major cancer hotspot region alters fatty acid metabolism in several cancer types [84]. FADS2 operates on at least 16 substrates, catalyzing the biosynthesis of several unsaturated fatty acids by Δ6, Δ8, and Δ4 desaturation [85]. The availability of fatty acid substrates for FADS2 in any tissue is the major determinant of the final product mixture for further membrane lipid synthesis and/or synthesis of signaling molecules [43,86,87]. *FASN*, *ELOVL6*, and *SCD* comprise the classical de novo FAS pathway and produce ω9 oleic acid (OA) (Figure 1b). These three genes were also among the most upregulated U/FAS genes, which is consistent with high OA production (Figure 4a,c). Indeed, the actual amount of OA produced is likely even higher than observed given that OA is further processed to downstream ω9 fatty acids, such as Mead acid and its derivative, DTrA (Figure 1b and Figure 4c).

Selective pressures may also explain the decreased ω3 and ω6 as well as elevated ω9 and fatty acid levels observed in NB (Figure 5a–c). Multiple studies have shown that ω3 HUFAs EPA and DHA have anticancer activity through the induction of apoptosis in multiple cancers (Figure 1b; Appendix A) [33,34,35]. DHA treatment of ovarian and lung cancer cell lines and an in vivo mouse model of ovarian cancer increased ROS production, induced apoptosis in vitro, and reduced both Ki67 staining and tumor size in vivo [88,89]. Interestingly, pre-treatment with the antioxidant NAC reversed these effects, suggesting that ROS production may be a key mechanism of growth inhibition by DHA. DHA and other HUFAs are known to be highly sensitive to ROS, which attack double bonds in fatty acid chains. ROS/HUFA reactions produce hydroperoxides on fatty acid chains, resulting in chain reactions that further propagate ROS production and damage [90]. These observations and our own provide a plausible rationale for gaining a selective advantage by suppressing ω3 synthesis in NB.

Chronic inflammation is associated with increased cancer incidence, whereas an inflammatory tumor microenvironment and acute inflammation have been shown to increase anti-cancer immune function and immune therapy efficacy [91]. While some adult tumors, such as melanoma, are considered “hot” tumors with high local inflammation and immune cell presence, pediatric solid tumors are most often “cold” [92]. NB is a classic cold tumor recently reported to have the lowest inflammatory response signature of 37 tumor types, suggesting that NB strongly suppresses inflammatory signaling. *MNA* disease has further reduced inflammatory signaling, which is consistent with the limited efficacy of immune therapies in NB [93].

Our observations regarding ω6 fatty acids provide a plausible explanation for the reduced inflammatory signature in NB. Both LA and ARA serve as key precursors of pro-inflammatory eicosanoids (Figure 1b and Figure 5a) [29]. Lower levels of both LA and ARA in NB reduce the pool of precursors available for production. We further observed reduced expression of all three (*LOX*, *COX*, and *Cytochrome P450*) enzyme pathways that produce inflammatory eicosanoids from ARA in high-risk NB and found that low expression of these genes was associated with higher risk of death (Figure 6b), and further that MYCN may mediate downregulation of most of them (namely *ALOX5*, *COX2*, *EPHX2*, and *CYP4V2*). Our observations are consistent with NB’s low inflammation signature, providing a plausible model for metabolic reprogramming that favors ω9 fatty acid production over ω6 ARA production. Recent studies have also demonstrated the pro-apoptotic effect of ARA on cancer cells and revealed the anti-tumor effects of ARA metabolism, providing further support for selective pressure to minimize both ARA and downstream eicosanoid production [94,95,96,97].

These data suggest that MYCN and MYC regulate the genes in the U/FAS pathway, identifying novel roles for MYC/N transactivation of *FADS2* and *FADS1*, with *ELOVL6* as an additional novel target of MYCN. Further, KEGG analysis of *MNA* vs. *MNon* NB patients showed that several fatty acid pathways were significantly enriched in the *MNA* group (Figure 7c). Half of HR NB patients were *MNA* and had a significantly enriched FAS pathway. Among *MNA* patients, KEGG pathways for FAS, fatty acid elongation, fat digestion/absorption, and fatty acid degradation were all significantly elevated, further supporting MYCN regulation of U/FAS metabolism in NB. MiRNA host genes *miR-22hg*, *miR-24hg*, and *miR-101hg* appeared to be suppressed by both MYCN and MYC modulation (Figure 7d,e), providing a possible mechanism for TSmiR suppression in NB and perhaps other MYC-driven cancers. It is also possible that U/FAS gene expression levels in cancer are, in part, regulated indirectly by MYC and MYCN through transcriptional suppression of TSmiR host gene expression and consequent release of U/FAS mRNAs from post-transcriptional TSmiR-mediated degradation. It is critical to note that it is likely that both *MYCN* and *MYC* modulate microRNA expression and function and are themselves sensitive to microRNA regulation. The interplay between NB risk class, *MYCN* status, and microRNA function represents a means of upregulating these genes in both *MNA* and *MNon* HR NB. This model thus establishes a novel regulatory network between *MYC*/*N*, U/FAS core genes, and multiple TSmiRs (which also target *MYCN* mRNA in NB).

## 5. Conclusions

*MNA* and TSmiR disruption are key players in NB pathology. Here, we endeavored to understand the role of U/FAS in pediatric NB. We showed that broad U/FAS disruption in HR NB by both MYCN activity and TSmiR disruption contribute to worse patient survival. We also reported altered lipid profiles in NB cells via decreased ω3 and ω6 and increased ω9 fatty acid levels, which may be selected to minimize ω3-driven ROS toxicity and ω6-driven pro-inflammatory signaling. Our results suggest an extended model of metabolic reprogramming, including UFAS, that may be an extension of the Warburg effect. We propose a model in which MYCN and TSmiRs oppositely regulate UFAS in NB as an extension of the MYC/N-driven Warburg effect, in part via increased production of acetyl-CoA, the fundamental building block of de novo FAS. This model introduces the UFAS genes *ELOLV6*, *FADS2*, and *FADS1* as novel targets of MYCN in NB and post-transcriptional regulation of U/FAS genes by multiple TSmiRs (Figure 2 and Figure 8). We propose a model in which *MYCN* and TSmiRs regulate UFAS and play an important role in NB, with implications for other *MYC* family-driven cancers. This model incorporates a novel complex relationship between *MYC/N* and TSmiRs that allows for combinatorial and/or independent function of these two powerful regulators of gene expression, which can contribute to U/FAS gene disruption in both *MNA* and *MNon* HR disease. This work connects disrupted U/FAS to poor survival in NB, providing a better understanding of how fatty acids may contribute to NB development and progression, as well as potential new considerations for treatment.

## Figures and Tables

**Figure 1 cancers-16-01590-f001:**
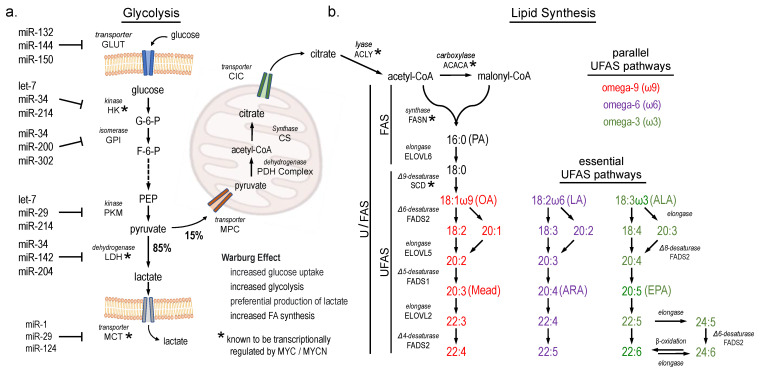
Glycolysis and fatty acid synthesis pathways. (**a**) Schematic of glycolysis genes and known microRNA regulators. Enzymes regulated by the MYC family are denoted with an asterisk (*). (**b**) Schematic of fatty acid synthesis. FASN catalyzes de novo FAS from acetyl-CoA and malonyl-CoA subunits up to sixteen unsaturated carbon chain palmitic acid (16:0, PA), which then enters elongation and desaturation pathways to yield oleic acid (18:1ω9, OA), which can continue along the de novo pathway to Mead acid (20:3ω9) and its elongation product, DTrA (22:3ω9). Essential ω6 linoleic acid (18:2ω6, LA) and alpha linolenic acid (18:3ω3, ALA) are elongated and desaturated by the same set of enzymes.

**Figure 2 cancers-16-01590-f002:**
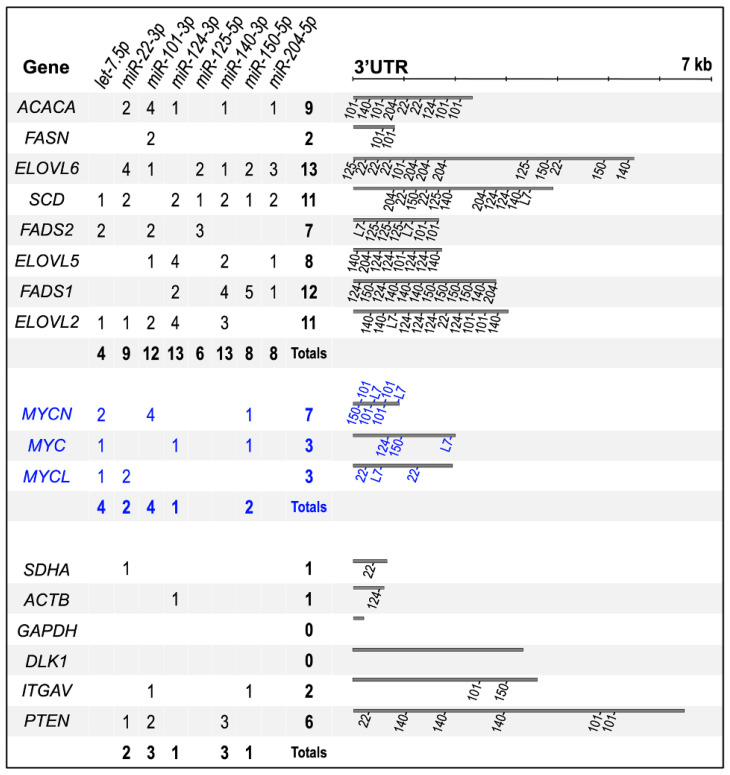
U/FAS gene 3′ UTRs showing positions of predicted microRNA target sites. The graphic shows the approximate locations of predicted miRNA target sites within 3′ UTRs of listed genes. Scale for 3′ UTR length located above UTRs. UTR lengths based on RefSeq sequences for transcript variant 1 in all cases. Numeric table shows the number of sites in multiple U/FAS genes, the MYC family (blue), several housekeeping genes (SDHA, ACTB, and GAPDH), and three unrelated genes (DLK1, ITGAV, and PTEN). Binding sites were identified using TargetScan Release 8.0.

**Figure 3 cancers-16-01590-f003:**
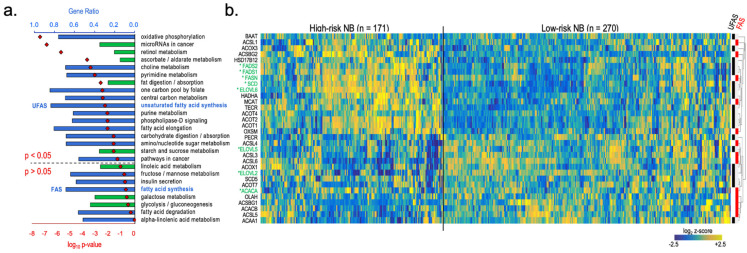
Comparative metabolic KEGG pathway analysis of high–and low–risk neuroblastoma patients. (**a**) KEGG pathway analysis of enriched metabolic or cancer-related pathways in HR (*n* = 171) over LR (*n* = 270) disease NB patients, stages 1–4. Blue bars are overrepresented pathways; green bars are underrepresented pathways; red diamonds are log_10_ *p*-values. Over- and underrepresented KEGG pathways were determined by Welch’s *t*-test followed by B-H FDR correction. (*GEO accession GSE62564*). (**b**) Heatmap of UFAS and FAS pathway gene expression in stage 1–4 neuroblastoma patients, comparing HR (*n* = 171) and LR disease (*n* = 270). U/FAS pathway genes from Figure 1b are titled in green and contain an asterisk (*). Genes associated with FAS and UFAS gene sets are identified by clustered red and black bars, respectively. Expression data are presented as log_2_ z-scores, quantifying standard deviations above (yellow) or below (blue) average expression among all patients.

**Figure 4 cancers-16-01590-f004:**
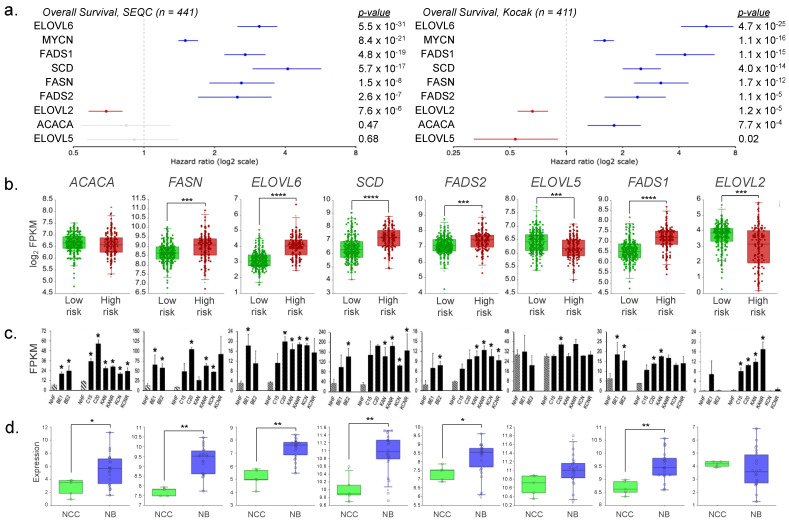
U/FAS gene expression is associated with worse overall survival in neuroblastoma. (**a**) Cox regression analysis of overall survival with U/FAS gene expression. Hazard ratios represent the relative risk of death associated with increased gene expression. Blue and red hazard ratios bars indicate elevated and decreased risk, respectively (*GEO accessions GSE62564*, *GSE45547*). (**b**) Gene expression comparison between NB patients with LR (*n* = 270) and HR (*n* = 171) disease. Significance was determined by one-way ANOVA (*GEO accession GSE62564*). (**c**) FPKM gene expression levels of U/FAS genes in NHF and human NB cell lines. Left set (NHF, BE1, BE2) NB cell lines represent *n* = 3 sequencing rounds; right set represents *n* = 2 rounds. Significance determined by *t*-test, * *p* < 0.05. (**d**) Expression comparison between human neural crest cells (NCC) and neuroblastoma cell lines (NB) (*GEO accessions GSE28019*, *GSE14340*). * *p* < 0.05, ** *p* < 0.001, *** *p* < 1 × 10^−6^, **** *p* < 1 × 10^−12^.

**Figure 5 cancers-16-01590-f005:**
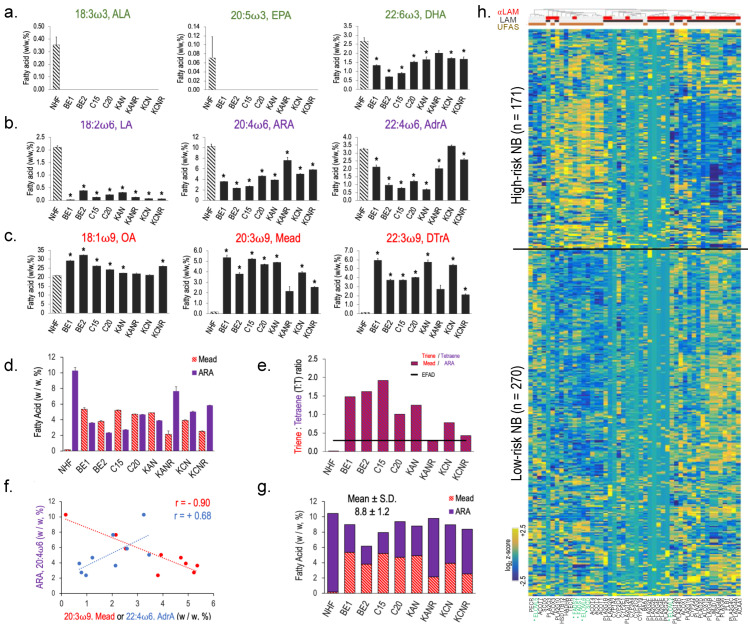
Omega-9 unsaturated fatty acids are elevated in neuroblastoma. (**a**–**c**) Fatty acid levels in NHF (striped bars) and human NB cell lines, presented as percent of overall lipid weight (*w*/*w*, %). Pathway precursors and key HUFAs are shown. Omega-3 fatty acids are titled in green (**a**), ω6 are titled in purple (**b**), and ω9 fatty acids are titled in red (**c**). (**d**) Comparison of Mead acid (red striped bars) and ARA (purple bars) levels in NHF and human NB cell lines. (**e**) T:T ratios in human cell lines. Comparison of Mead acid-ARA ratios for NHF and human NB cell lines. Essential fatty acid deficiency (EFAD) is defined by a T:T ratio above 0.25 (denoted by black line). (**f**) Correlation graphs of ARA levels to levels of Mead acid (red) or ARA-elongation product, AdrA (blue). (**g**) Sum of Mead acid (red strips) and ARA (purple) levels across NHF and human NB cell lines. The mean and SD of all cell lines is 8.8 ± 1.2 (*w*/*w*, %). (**h**) Heatmap of α-linolenic acid metabolism (αLAM), linoleic acid metabolism (LAM), and unsaturated fatty acid synthesis (UFAS) gene expression, comparing high-risk (*n* = 171) and low-risk (*n* = 270) disease in NB patients, stages 1–4. U/FAS pathway genes from Figure 1b are titled in green and contain asterisks (*). Genes associated with αLAM, LAM, and U/FAS gene sets are identified by clustered red, black, and brown bars, respectively. Expression data is presented as normalized values (log_2_ z-scores).

**Figure 6 cancers-16-01590-f006:**
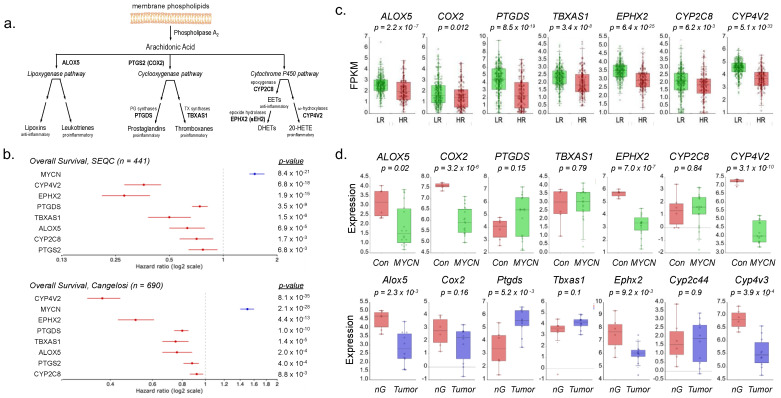
ARA metabolism pathways are downregulated in high-risk neuroblastoma. (**a**) Schematic of ARA immune-active metabolism. ARA, 20:4ω6, is metabolized by the lipoxygenase, cyclooxygenase, and cytochrome P450 enzymatic pathways to produce various immune-active eicosanoids. (**b**) Cox regression analysis of overall survival with U/FAS gene expression. Hazard ratios represent the relative risk of death associated with increased gene expression. Blue and red hazard ratios bars indicate elevated and decreased risk, respectively (*GEO accessions GSE62564*, *GSE120572*). (**c**) Associated gene expression comparison between NB patients (stages 1–4) with LR (*n* = 270) and HR (*n* = 171) disease by RNA-sequencing FPKM values. Differential expression significance was determined by one-way ANOVA (*GEO accession GSE62564*). (**d**) ARA metabolizing gene expression in response to MYCN, by microarray. Upper panel: SK-N-AS cells either non-induced (Con, *n* = 6) or induced (*MYCN*, *n* = 18) to express *MYCN*. Lower panel: normal ganglia (nG, *n* = 6) compared to TH-MYCN-driven tumors (Tumor, *n* = 10). *Cyp2c44* and *Cyp4v3* are murine homologs of *CYP2C8* and *CYP4V2*, respectively. Differential expression significance was determined by one-way ANOVA (GEO accessions *GSE16478*, *GSE43419*).

**Figure 7 cancers-16-01590-f007:**
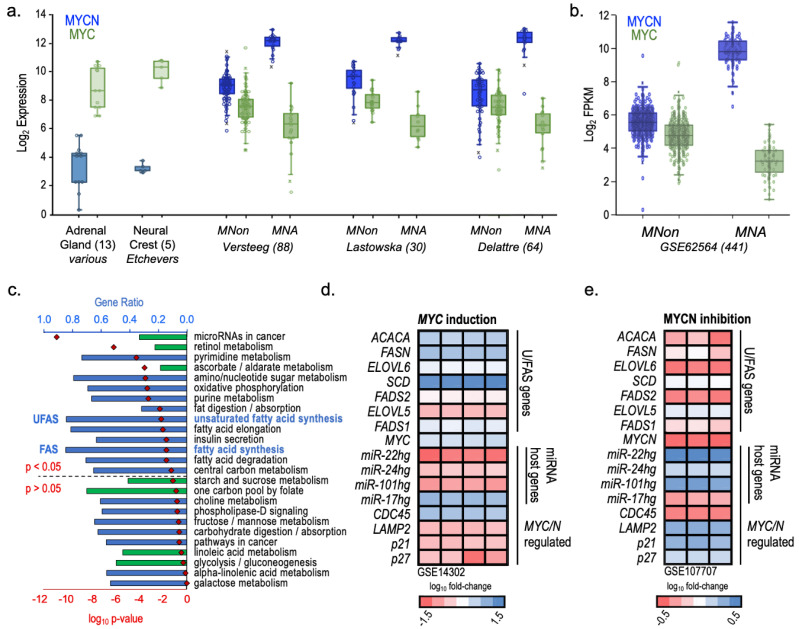
Expression patterns of MYC/N and related genes in NB by KEGG Gene Set Analysis. (**a**) Microarray expression of *MYC* and *MYCN* in *MNA* and *MNon* patients from three different NB data sets and normal adrenal gland (NB tissue of origin) and trunk neural crest cells (NB cells of origin). (**b**) Gene expression comparisons in log_2_ FPKM of *MYC* and *MYCN* in *MNA* (*n* = 88) and *MNon* (*n* = 353) NB patients. (**c**) KEGG pathway analysis of enriched metabolic or cancer-related pathways in *MNA* (*n* = 88) over *MNon* (*n* = 353) stage 1–4 NB patients. Blue bars are overrepresented pathways; green bars are underrepresented pathways; red diamonds are log_10_ *p*-values. Over- and underrepresented KEGG pathways were determined by Welch’s *t*-test followed by B-H FDR correction. (**d**,**e**) Heatmaps of U/FAS and MYC/N target gene expression after *MYC* induction (**d**) or MYCN inhibition by JQ-1 bromo-domain inhibitor (**e**). Each column represents a replicate experiment. (Figure 6c: *GEO accession GSE62564Heatmaps of U/FAS and MYC/N target gene*).

**Figure 8 cancers-16-01590-f008:**
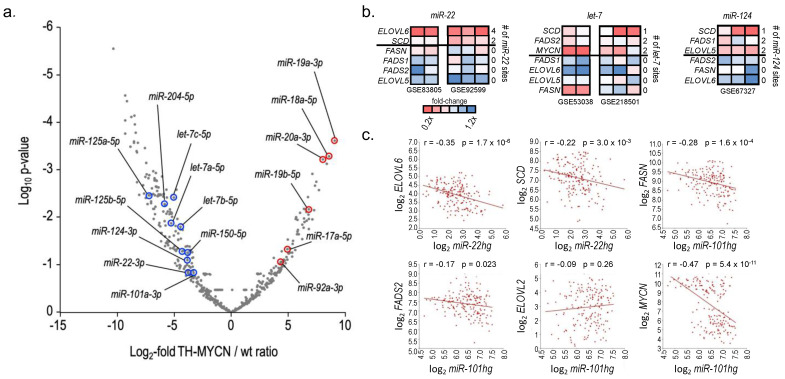
Tumor suppressor miRNAs are downregulated in *TH-MYCN*-driven neuroblastoma and result in reduced expression of their U/FAS mRNA targets. (**a**) Volcano plot of differentially expressed miRNAs in *TH-MYCN* NB tumors (*n* = 12) and wildtype (wt) ganglia (*n* = 12; NB tissue-of-origin). The *x*-axis indicates log_2_-fold change between tumors and ganglia; the *y*-axis indicates log_10_ *p*-values; blue circles signify tumor suppressor miRNAs from Figure 2; red circles signify miRNAs of the miR-17-92 oncomiR cluster (*r2 dataset identifier E-MTAB-2618*). (**b**) Heatmaps of U/FAS gene expression changes in response to *miR-22* (left), *let-7* (middle), and *miR-124* (right) overexpression. Each heat map is a unique study labeled with its GEO accession number. Each column represents a replicate experiment. Number of miRNA sites identified in the 3′ UTR of the listed gene are listed along the right of each miRNA set. (**c**) Correlation plots for mRNA-target gene expression vs. miRNA-host gene for *miR-22* or *miR-101* in HR NB patients (*n* = 176) (Figure 6c: *GEO accession GSE62564*).

## Data Availability

The RNA-Seq dataset generated in this study will be deposited in NCBI’s Gene Expression Omnibus (GEO) and is accessible through GEO reference series (GSE) (to be deposited (http://www.ncbi.nlm.nih.gov/geo/query/acc.cgi?acc=GSE(TBD)). Additional data generated in this study are available within the article and its Appendix A. Details for GEO accession datasets used for secondary analyses are given in Appendix A.

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
