# Peer review of "Unsaturated Fatty Acid Synthesis Is Associated with Worse Survival and Is Differentially Regulated by MYCN and Tumor Suppressor microRNAs in Neuroblastoma"

_cancers, 2024, doi:10.3390/cancers16081590_

Round 1
Reviewer 1 Report (Previous Reviewer 1)
Comments and Suggestions for Authors
The International Neuroblastoma Pathology Classification (INPC)1 uses age, neuroblastic maturation, Schwannian stromal content, and MKI as prognostic indicators. Unfavorable indicators include undifferentiated neuroblastoma and high MKI.
The manuscript does not provide information about favorable and unfavorable Histology groups/International Neuroblastoma Pathology Prognostic Classification. Please see attachment for additional information.
Poor prognosis is not a relative term. Please see attachment for additional information.
The manuscript should be revised to include additional information about favorable and unfavorable Histology groups/International Neuroblastoma Pathology Prognostic Classification, according to the current AJCC/CAP guidelines.

Author Response
Comments and Suggestions for Authors:
- The International Neuroblastoma Pathology Classification (INPC)1 uses age, neuroblastic maturation, Schwannian stromal content, and MKI as prognostic indicators. Unfavorable indicators include undifferentiated neuroblastoma and high MKI.
Combining this comment with the second and fourth comments as well as the informative attachment you provided, we have written a new section into the beginning of the introduction that addresses this and the other comments (lines 2-81). Please see our response to comment #2 for our full response.
- The manuscript does not provide information about favorable and unfavorable Histology groups/International Neuroblastoma Pathology Prognostic Classification. Please see attachment for additional information.
We are grateful for the additional information you provided. Combining the information in the provided Protocol for the Examination of Resection Specimens From Patients With Neuroblastoma as well as the first, second, and fourth comments from your review, we have written a new section into the beginning of the introduction section to appropriately address these comments. We believe that properly addressing your concerns has allowed us to present an appropriate framework upon which to overlay the importance of our observations in unfavorable histology groups, and how those groups relate to high-risk disease and MYCN amplification. We believe the end result of this is a significantly approved framing of this work.
- Poor prognosis is not a relative term. Please see attachment for additional information.
Thank you for this follow up comment, and for the additional information you included. We have limited the use of prognosis to the new section of the introduction, where we put the term in the context of major events in neuroblastoma such as MYCN amplification. Throughout the manuscript, we discuss patient outcomes in terms of overall survival instead of in prognostic terms and comment on whether a given group has better or worse overall survival. We feel this presentation is most in line with the message of the manuscript. In addition, as per our last revision, we have shifted our analysis of gene expression and patient survival to Cox regression and Hazard Ratios in the main figures (Figure 3a and 5b).
- The manuscript should be revised to include additional information about favorable and unfavorable Histology groups/International Neuroblastoma Pathology Prognostic Classification, according to the current AJCC/CAP guidelines.
Combining this comment with the first and second comments as well as the informative attachment you provided, we have written a new section in the beginning of the introduction that should address this and the other comments. Please see our response to comment #2 for our full response.
Reviewer 2 Report (Previous Reviewer 2)
Comments and Suggestions for Authors
This study highlights the impact of unsaturated fatty acid synthesis (UFAS) on neuroblastoma (NB) pathogenesis, particularly in cases with MYCN amplification (MNA). It finds that omega-3, omega-6, and omega-9 fatty acid synthesis pathways, along with key enzymes (FASN, ELOVL6, SCD, FADS2, FADS1), are disrupted in high-risk NB, linking to poor survival. RNA-Seq and fatty acid profiling connect MYCN overexpression with altered fatty acid metabolism, influenced by tumor suppressor microRNAs (TSmiRs). The data indicate a unique fatty acid pathway in NB, with a preference for omega-9 synthesis, potentially driving the aggressiveness in high-risk NB. This study presents intriguing findings, and the manuscript is well-crafted. However, there are several questions that need to be addressed to further strengthen the paper before considering it for publication.
Minor Comments:
1. For the Materials and Methods of RNA-seq, please include more details, such as, how many reads were sequenced, pair-end, or single-end seq, how the sequencing data was normalized, and whether spike-in was used et al.
2. The format of Table 1 needed to be adjusted, the fonts were overlapped with each other.
3. Line 194, it’s unclear what standard the author used to define “HR” and “LR” NB patients. The same as line 226 “eight human NB cell lines derived from high-risk patient”. Does line 194 and line 226 use the same standard to define “High risk”? Without this information, it would be hard to interpret whether this grouping is reasonable and whether the following analysis is meaningful.
4. C15 and C20 cell lines exhibit a non-amplified MYCN genetic background and are classified as 'High risk.' Notably, certain U/FAS genes show upregulation in these lines (see Figure 3c), challenging the hypothesis that MYCN acts as a positive regulator of U/FAS gene expression. This observation suggests that U/FAS gene expression may be more closely associated with the 'High risk' classification. Consequently, this raises a critical question, echoing the inquiry in point 3: what defines 'High risk' in this context?
5. If comparing MYCN non-amplified NB vs MYCN amplified NB cell lines, the correlation of MYCN and U/FAS gene expression is less significant based on current RNA-seq data.
6. Figure 3c, SCD, C20 group looks significantly increased, please verify.
7. Line 375, this sentence is confusing, “bromo-domain MYC/N inhibitor JQ-1”. MYC/N does not have a bromo domain.
Author Response
Comments and Suggestions for Authors:
This study highlights the impact of unsaturated fatty acid synthesis (UFAS) on neuroblastoma (NB) pathogenesis, particularly in cases with MYCN amplification (MNA). It finds that omega-3, omega-6, and omega-9 fatty acid synthesis pathways, along with key enzymes (FASN, ELOVL6, SCD, FADS2, FADS1), are disrupted in high-risk NB, linking to poor survival. RNA-Seq and fatty acid profiling connect MYCN overexpression with altered fatty acid metabolism, influenced by tumor suppressor microRNAs (TSmiRs). The data indicate a unique fatty acid pathway in NB, with a preference for omega-9 synthesis, potentially driving the aggressiveness in high-risk NB. This study presents intriguing findings, and the manuscript is well-crafted. However, there are several questions that need to be addressed to further strengthen the paper before considering it for publication.
Minor Comments:
- For the Materials and Methods of RNA-seq, please include more details, such as, how many reads were sequenced, pair-end, or single-end seq, how the sequencing data was normalized, and whether spike-in was used et al.
We have edited the materials and methods to include this information (lines 161-163).
- The format of Table 1 needed to be adjusted, the fonts were overlapped with each other.
This has been corrected. The fonts in Table 1 should no longer be overlapped.
- Line 231, it’s unclear what standard the author used to define “HR” and “LR” NB patients. The same as line 261 “eight human NB cell lines derived from high-risk patient”. Does line 231 and line 261 use the same standard to define “High risk”? Without this information, it would be hard to interpret whether this grouping is reasonable and whether the following analysis is meaningful.
We appreciate this comment. In response to this and others, we have added a new section to the introduction where we go into detail about risk classification, MYCN amplification, and the relationship between the two (lines 2-81). Also, we discuss in the text that while MYCN non-amplified NB have considerably less MYCN, they do have a degree of compensatory c-MYC expression (Figure 6a), which when considered in total, supports the idea that even MYCN non-amplified cells retain MYC/N signaling. Also, we have generated a new Supplemental Figure S3 which shows that CHLA15 and CHLA20 have elevated c-MYC levels that are similar to MYCN levels in KAN and KCN cell pairs, further supporting this idea. We have also added additional text to this section to better incorporate these observations into the conclusions of the manuscript.
Also, while we believe the cell lines in Figure 3c are very likely from high-risk patients, we do not have access to clinical data that can support our use of this term with them in the manuscript. For this reason, we have removed “derived from high-risk patients” and rewritten that sentence as: “We also examined the expression patterns of this gene set in eight human NB cell lines pairs derived from tumors of the same patient at diagnosis (BE1, CHLA15, KAN, KCN) and post-relapse (BE2, CHLA20, KANR, KCNR) using RNA sequencing (RNA-Seq) analysis (Supplemental Table S2)”; (lines 265-269). This relegates Figure 3C to showing that in established cell lines, our genes of interest, FASN, SCD, ELOVL6, FADS2, and FADS1 are upregulated in NB over normal cells. We go a step further in Figure 3d, where we compare NB cell lines to the NB cell of origin, neural crest cells (NCC), where we see the same pattern of upregulation of the U/FAS genes in NB over NCC.
- C15 and C20 cell lines exhibit a non-amplified MYCN genetic background and are classified as 'High risk.' Notably, certain U/FAS genes show upregulation in these lines (see Figure 3c), challenging the hypothesis that MYCN acts as a positive regulator of U/FAS gene expression. This observation suggests that U/FAS gene expression may be more closely associated with the 'High risk' classification. Consequently, this raises a critical question, echoing the inquiry in point 3: what defines 'High risk' in this context?
Please see our response to comment #3 as we believe this comment may be mostly addressed there. In addition, we believe the new section written into the introduction may address this concern as well. Briefly, our initial observation in patients was in comparison to high-risk and low-risk patients. We then explored contribution of both MYCN and c-MYC to regulation of these genes (Figure 6) as well as tumor suppressor microRNAs (Table 1, Figure 7). Our interpretation of our collective observations is that both MYC/N and tumor suppressor microRNAs contribute to the regulation of UFAS genes in both NB in general and more so in high-risk disease (both MYCN amplified and non-amplified). We concede that there may be other regulatory mechanisms at play. Addressing this is the current focus of our follow up studies. We have also added some language to the discussion and conclusion to better address this complexity (lines 584-590, lines 606-608).
- If comparing MYCN non-amplified NB vs MYCN amplified NB cell lines, the correlation of MYCN and U/FAS gene expression is less significant based on current RNA-seq data.
Please see our response to comment #3 as we believe this comment may be addressed there.
- Figure 3c, SCD, C20 group looks significantly increased, please verify.
Thank you for pointing this out. You are correct. C20 is significantly increased for SCD expression, whereas KCNR is not. The stars for significance in that panel were incorrectly shifted to the right by one. This has now been corrected and 3c should be correct for significance for all eight genes.
- Line 375, this sentence is confusing, “bromo-domain MYC/N inhibitor JQ-1”. MYC/N does not have a bromo domain.
We have edited this sentence to correct for the confusion and ensure accuracy.
Round 2
Reviewer 1 Report (Previous Reviewer 1)
Comments and Suggestions for Authors
The revised manuscript looks good.
This manuscript is a resubmission of an earlier submission. The following is a list of the peer review reports and author responses from that submission.
Round 1
Reviewer 1 Report
Comments and Suggestions for Authors
The manuscript covers a potentially very interesting subject, but suffers from serious flaws:
1. The International Neuroblastoma Pathology Classification (INPC) uses age, neuroblastic maturation, Schwannian stromal content, and MKI as prognostic indicators; however, no information is provided by the authors about favorable and unfavorable histology groups (neuroblastic maturation, Schwannian stromal content, and MKI as prognostic indicators).
2. How was the poor prognosis established? Please provide additional information.
3. Comparative genomic hybridization is typically used to evaluate for segmental chromosomal aberrations (especially 1p deletion, 11q deletion, and/or 17q gain), which are associated with high-risk tumors, whereas alterations in the numbers of whole chromosomes are associated with lower risk tumors. The manuscript does not provide any information about the chromosomal aberrations mentioned above.
Reviewer 2 Report
Comments and Suggestions for Authors
The research by Dennis A. Sheeter and colleagues seeks to explore the potential link between MYCN, microRNA, and the Unsaturated fatty acid synthesis pathway gene expression in neuroblastoma. The background introduction is adequate, and the hypothesis is both intriguing and plausible. However, the experiment design, the manuscript writing, and the figure presentation are a mess.
Here are a few comments:
Major comments:
1. In lines 207-210 and Figure 2a. It's unclear why this particular dataset was chosen and how patients were categorized into high-risk and low-risk groups. Additionally, it's ambiguous whether the KEGG pathway enrichment indicates a "high-risk" over "low-risk" comparison or vice versa.
2. Figure 2b is quite challenging to comprehend. Firstly, comparing gene expression levels between two distinct heatmaps is unreliable. Secondly, the z-score in this figure is used to compare the expression levels of various genes within the same sample, which is not meaningful.
3. Employing Kaplan scanning to segregate high and low groups as depicted in Figure 3b is not reasonable. The author should categorize the Kaplan-Meier groups as previously described, into "high-risk, 171" and "low-risk, 270" groups.
4. Figure 3c. Figure 4a,b,c,d,e,g. Normal human fibroblast cells are derived from connective tissue. Whereas Neuroblastoma cells originate from neuroblasts, which are immature nerve cells. It’s unreasonable to compare the expression level of Neuroblastoma cells to NHF.
5. It is quite unclear which gene sets are represented by the terms FAS, de novo FAS, UFAS, U/FAS, and U/FAS-core. Interpreting the data is impossible without additional clarification.
6. Line 250, I didn’t see any of these tables “Supplemental Tables S1, S2, and S4”.
Minor comments:
7. Line 53 “Several genes in this pathway are known to be transcriptionally regulated by MYC”, the term “transcriptionally regulated” is ambiguous, please clarify whether that means activation, repression, or others.
8. Line 70 “regulate mRNAs post-transcriptionally”, the term “regulate” is ambiguous, please clarify whether that means activation, repression, or others.
Comments on the Quality of English LanguageNot a big issue.